# Anytime-valid, Bayes-assisted, Prediction-Powered Inference

**Valentin Kilian**\*
Department of Statistics,
University of Oxford
kilian@stats.ox.ac.uk

**Stefano Cortinovis**\*
Department of Statistics,
University of Oxford
cortinovis@stats.ox.ac.uk

**François Caron**
Department of Statistics,
University of Oxford
caron@stats.ox.ac.uk

## Abstract

Given a large pool of unlabelled data and a smaller amount of labels, prediction-powered inference (PPI) leverages machine learning predictions to increase the statistical efficiency of confidence interval procedures based solely on labelled data, while preserving fixed-time validity. In this paper, we extend the PPI framework to the sequential setting, where labelled and unlabelled datasets grow over time. Exploiting Ville's inequality and the method of mixtures, we propose prediction-powered confidence sequence procedures that are asymptotically valid uniformly over time and naturally accommodate prior knowledge on the quality of the predictions to further boost efficiency. We carefully illustrate the design choices behind our method and demonstrate its effectiveness in real and synthetic examples.

## 1 Introduction

Increasing the sample size of an experiment is arguably the single simplest way to improve the precision of the statistical conclusions drawn from it. However, in many fields – such as healthcare, finance, and social sciences – obtaining labelled data is often costly and time-consuming. In these settings, using machine learning (ML) models to impute additional labels represents a tempting alternative to expensive data collection, albeit at the risk of introducing bias. Prediction-powered inference (PPI) [1] is a recently introduced framework for valid statistical inference in the presence of a small labelled dataset and a large number of unlabelled examples paired with predictions from a black-box model.

Formally, given an input/output pair $(X, Y) \sim \mathbb{P} = \mathbb{P}_X \times \mathbb{P}_{Y|X}$, consider the goal of estimating

$$\theta^\star = \arg\min_{\theta \in \mathbb{R}} \ \mathbb{E}[\ell_\theta(X, Y)], \tag{1}$$

where $\ell_\theta(x, y)$ is a convex loss function parameterised by $\theta \in \mathbb{R}$. As an example, the mean $\theta^\star = \mathbb{E}[Y]$ is the estimand induced by the squared loss $\ell_\theta(x, y) = (\theta - y)^2/2$. For $t = 1, 2, \ldots$, we observe a sequence of independent random variables $Z_t$, either drawn from $\mathbb{P}$ (labelled sample) or from $\mathbb{P}_X$ (unlabelled sample), and we are provided with a black-box prediction rule $f$ that maps any input $x$ to a prediction $f(x)$.

Let $(X_i, Y_i)_{i \geq 1}$ and $(\widetilde{X}_j)_{j \geq 1}$ denote the subsequences of labelled and unlabelled samples, respectively. For $n = 1, 2, \ldots$, let $N_n$ denote the number of unlabelled samples observed before the $n$th labelled one, and assume that $N_n \geq n$, with $N_n \gg n$ in typical settings. PPI constructs an (asymptotic) $1 - \alpha$ confidence interval (CI) $\mathcal{C}_{\alpha,n}^{\mathrm{pp}}$ for $\theta^\star$, that exploits the auxiliary information encoded in $f$. To this end, under mild assumptions, $\theta^\star$ can be expressed as the solution to

$$g_{\theta^\star} := \mathbb{E}[\ell'_{\theta^\star}(X, Y)] = 0, \tag{2}$$

---

\*Equal contribution. Order decided by coin toss.

where $\ell'_\theta$ is a subgradient of $\ell_\theta$ with respect to $\theta$. The quantity $g_\theta$ in Equation (2) can be decomposed as $g_\theta = m_\theta + \Delta_\theta$, where

$$m_\theta := \mathbb{E}[\ell'_\theta(X, f(X))] \quad \text{and} \quad \Delta_\theta := \mathbb{E}[\ell'_\theta(X, Y) - \ell'_\theta(X, f(X))], \tag{3}$$

where $m_\theta$ represents a measure of fit of the predictor, while $\Delta_\theta$, the *rectifier*, accounts for the discrepancy between the predicted outputs $f(X)$ and the true labels $Y$. If $\mathcal{C}^g_{\alpha,\theta,n}$ is a $(1-\alpha)$ confidence interval for $g_\theta$, then the PPI confidence interval $\mathcal{C}^{\mathrm{pp}}_{\alpha,n}$, defined as

$$\mathcal{C}^{\mathrm{pp}}_{\alpha,n} = \left\{ \theta \mid 0 \in \mathcal{C}^g_{\alpha,\theta,n} \right\}, \tag{4}$$

also achieves the desired coverage, i.e., $\Pr(\theta^\star \in \mathcal{C}^{\mathrm{pp}}_{\alpha,n}) \geq 1 - \alpha$. Constructing $\mathcal{C}^g_{\alpha,\theta,n}$ relies on estimating $g_\theta$, for which PPI defines an estimator leveraging both the unlabelled data and the prediction rule $f$. The resulting method outperforms standard CI procedures based on the labelled data alone when $f$ is sufficiently accurate and $N_n \gg n$. Intuitively, this is because, in this case, $\Delta_\theta$ is close to zero, while $m_\theta$ can be estimated with low variance from the unlabelled data.

Crucially, coverage of the PPI CI (4) is guaranteed only at a fixed time, i.e., for a labelled sample size $n$ fixed in advance. This is undesirable in many practical settings – such as online learning, real-time monitoring, or sequential decision-making – where it is essential to continuously draw conclusions as new data arrive. In this work, we address this by proposing an *anytime-valid* extension of the PPI CI (4). That is, we define a confidence sequence $(\mathcal{C}^{\mathrm{avpp}}_{\alpha,n})_{n \geq 1}$, satisfying asymptotically the stronger coverage guarantee

$$\Pr(\theta^\star \in \mathcal{C}^{\mathrm{avpp}}_{\alpha,n} \text{ for all } n \geq 1) \geq 1 - \alpha,$$

while still taking advantage of the prediction rule $f$. Analogously to standard PPI, we construct a confidence sequence $(\mathcal{C}^g_{\alpha,\theta,n})_{n \geq 1}$ for $g_\theta$ and define $\mathcal{C}^{\mathrm{avpp}}_{\alpha,n}$ through Equation (4) for $n \geq 1$. While our approach is agnostic to the specific form of the confidence sequence $(\mathcal{C}^g_{\alpha,\theta,n})_{n \geq 1}$, we mainly focus on asymptotic confidence sequences [2], as they provide a versatile time-uniform analogue of standard CLT-based CIs that applies to the PPI framework above in full generality. Moreover, being based on the method of mixtures [3, 4, 5], they can readily accommodate prior information on the quality of the prediction model $f$. In particular, by means of a zero-centred prior on the rectifier $\Delta_\theta$, we obtain tighter confidence sequences when the predictions are good, extending the fixed-time Bayes-assisted approach of Cortinovis and Caron [6].

The remainder of the paper is organised as follows. Section 2 reviews related work. Section 3 provides background on (asymptotic) confidence sequences and discusses how prior information may be incorporated into their construction. Section 4 presents PPI in the context of control-variate estimators, whose asymptotic properties are crucial for our approach to anytime-valid, Bayes-assisted PPI, which is described in Section 5. Section 6 demonstrates the benefits of our method on synthetic and real data. Finally, Section 7 discusses limitations of our approach and further extensions. Proofs and additional experiments are provided in the Supplementary Material.

## 2 Related Work

PPI was introduced by Angelopoulos et al. [1] as a general framework for valid statistical inference with black-box ML predictors, and later extended in Angelopoulos et al. [7]. Closely related ideas appear in the literature on semi-supervised inference, missing-data methods, survey sampling, and double machine learning [8, 9, 10, 11, 12]. More recently, Cortinovis and Caron [6] proposed a Bayes-assisted variant of PPI. All of these contributions target fixed-time confidence intervals.

Confidence sequences (CS) were introduced by Darling and Robbins [13] and further developed by Robbins and Siegmund [4] and Lai [5], building on earlier work by Ville [3] and Wald [14]. Interest has surged again in recent years [15, 2], motivated by applications such as A/B testing. The notion is closely linked to e-values [16, 17]. Building on the e-value framework and on earlier work by Zrnic and Candès [18] and Waudby-Smith and Ramdas [15], Csillag et al. [19] proposed an exact, time-uniform PPI method that yields CS under stronger conditions (e.g., existence of bounded e-values) but does not leverage prior knowledge about the quality of the ML predictions. Furthermore, applying their method requires an active-learning setup in which, at each time $t$, the observation $Z_t$ can be labelled with strictly positive probability. In particular, it is not applicable to deterministic sequences of observations, such as those describing a large initial pool of unlabelled data followed by a stream of labelled data, which are the main focus of our experiments.

In the setting of double machine learning and semiparametric inference, Dalal et al. [20] and Waudby-Smith et al. [2] derive asymptotic confidence sequences for target parameters in the presence of high-dimensional nuisance components.

# 3 Asymptotic (Bayes-assisted) confidence sequences

In this section, we first review background on (asymptotic) confidence sequences (CS), and then show how prior information can be incorporated into asymptotic CS procedures, leading to asymptotic Bayes-assisted confidence sequences.

## 3.1 Background

We start by defining an exact confidence sequence [13], a time-uniform analogue of classical CIs.

**Definition 1** (Confidence sequence). *Let $(\mathcal{C}_{\alpha,t})_{t \geq 1}$ be a sequence of random subsets of $\mathbb{R}$. For $\alpha \in (0,1)$, $(\mathcal{C}_{\alpha,t})_{t \geq 1}$ is a $1 - \alpha$ confidence sequence for a fixed parameter $\mu \in \mathbb{R}$ if*

$$\Pr(\mu \in \mathcal{C}_{\alpha,t} \text{ for all } t \geq 1) \geq 1 - \alpha. \tag{5}$$

We now introduce the notion of an asymptotic confidence sequence (AsympCS) [2, 20].

**Definition 2** (Asymptotic confidence sequence). *Let $\alpha \in (0,1)$ and $(a_t)_{t \geq 1}$ be a real sequence such that $\lim_{t \to \infty} a_t = 0$. Let $(\widehat{\mu}_t)_{t \geq 1}$ be a consistent sequence of estimators of $\mu$. The sequence of random intervals $(\mathcal{C}_{\alpha,t})_{t \geq 1}$, with $\mathcal{C}_{\alpha,t} = [\widehat{\mu}_t - L_t, \widehat{\mu}_t + U_t]$ and $L_t > 0$, $U_t > 0$, is said to be an asymptotic confidence sequence with (little-o) approximation rate $a_t$ if there exists a (usually unknown) confidence sequence $(\mathcal{C}^{\star}_{\alpha,t})_{t \geq 1}$, with $\mathcal{C}^{\star}_{\alpha,t} = [\widehat{\mu}_t - L^{\star}_t, \widehat{\mu}_t + U^{\star}_t]$, such that*

$$\Pr(\mu \in \mathcal{C}^{\star}_{\alpha,t} \text{ for all } t \geq 1) \geq 1 - \alpha$$

*and, almost surely as $t \to \infty$, $\max\{L^{\star}_t - L_t, U^{\star}_t - U_t\} = o(a_t)$.*

Thus, an asymptotic CS may be regarded as an approximation of an exact CS that becomes arbitrarily accurate in the limit. It is worth noting that, while classical fixed-sample asymptotic CIs rely on *convergence in distribution* of scaled estimators, asymptotic confidence sequences rely on *almost sure convergence at a given rate* of the centred lower and upper bounds relative to those of an underlying exact CS. The following is an example of an asymptotic CS that applies to i.i.d. data.

**Theorem 1.** *Let $(Y_t)_{t \geq 1}$ be a sequence of i.i.d. random variables with mean $\mu$ and such that $\mathbb{E}|Y_1|^{2+\delta} < \infty$ for some $\delta > 0$. For any $t \geq 1$, let $\overline{Y}_t$ be the sample mean, and $\widehat{\sigma}_t^2$ be the sample variance based on the first $t$ observations. For any parameter $\rho > 0$, the sequence of intervals defined as*

$$\mathcal{C}^{\texttt{NA}}_{\alpha,t}(\overline{Y}_t, \widehat{\sigma}_t; \rho) := \left[ \overline{Y}_t \pm \frac{\widehat{\sigma}_t}{\sqrt{t}} \sqrt{\left(1 + \frac{1}{t\rho^2}\right) \log\left(\frac{t\rho^2 + 1}{\alpha^2}\right)} \right] \tag{6}$$

*forms a $(1 - \alpha)$–AsympCS with approximation rate $1/\sqrt{t \log t}$ for $\mu$.*

For the sequel, it is useful to highlight some aspects of the proof of this theorem. First, if the random variables $(Y_t)_{t \geq 1}$ were Gaussian with variance $\sigma^2$, then $\mathcal{C}^{\texttt{NA}}_{\alpha,t}(\overline{Y}_t, \sigma; \rho)$ would form an exact CS. This follows from combining the method of mixtures for nonnegative martingales with Ville's inequality [3, 4, 5, 21]. Second, the proof relies on KMT strong coupling [22, 23]: there exists i.i.d. Gaussian random variables $(W_t)_{t \geq 1}$ with mean $\mu$ and variance $\text{var}(Y)$ such that

$$\frac{1}{t} \sum_{i=1}^{t} Y_i = \frac{1}{t} \sum_{i=1}^{t} W_i + o\left(\frac{1}{\sqrt{t \log t}}\right) \text{ a.s. as } t \to \infty.$$

Such a coupling plays a central role in constructing asymptotic confidence sequences, serving as a substitute for the CLT assumption underlying in classical fixed-sample CIs. The construction in Theorem 1 extends beyond the i.i.d. case, provided a similar coupling exists.

**Theorem 2.** *Let $(\widehat{\mu}_t)_{t \geq 1}$ be a consistent sequence of estimators of $\mu$. Assume that there exists a sequence of i.i.d. Gaussian random variables $(W_i)_{i \geq 1}$, with mean $\mu$ and variance $\sigma^2$, such that*

$$\widehat{\mu}_t = \frac{1}{t} \sum_{i=1}^{t} W_i + o\left(\frac{1}{\sqrt{t \log t}}\right) \text{ a.s. as } t \to \infty. \tag{7}$$

Let $(\widehat{\sigma}_t^2)_{t \geq 1}$ be a consistent sequence of estimators of $\sigma^2$ with $|\widehat{\sigma}_t - \sigma| = o\left(\frac{1}{\log t}\right)$ a.s. Then, for any parameter $\rho > 0$, the sequence of intervals $(\mathcal{C}_{\alpha,t}^{\mathtt{NA}}(\widehat{\mu}_t, \widehat{\sigma}_t; \rho))_{t \geq 1}$ forms a $(1-\alpha)$–AsympCS with approximation rate $1/\sqrt{t \log t}$ for $\mu$.

The asymptotic CS (6) includes a tuning parameter $\rho$, which can be chosen so as to minimise the width of the interval at a specified time $t$; see [2, Appendix B.2]. However, this method does not allow the incorporation of prior information about the parameter of interest to yield tighter intervals when the data align with such assumptions: the width of Equation (6) is independent of $\overline{Y}_t$.

## 3.2 Asymptotic Bayes-assisted confidence sequences

To address this, we introduce a Bayes-assisted analogue of Theorem 1.

**Theorem 3** (Bayes-assisted AsympCS – i.i.d. case). *Let $(Y_t)_{t \geq 1}$ be a sequence of i.i.d. random variables with unknown mean $\mu$ and unknown variance $\sigma^2$, and such that $\mathbb{E}|Y_1|^{2+\delta} < \infty$ for some $\delta > 0$. For any $t \geq 1$, let $\overline{Y}_t$ be the sample mean, and $\widehat{\sigma}_t^2$ be the sample variance based on the first $t$ observations. Let $\eta_t : \mathbb{R} \to (0, \sqrt{t/(2\pi)})$ be defined as*

$$\eta_t(z) = \int_{-\infty}^{\infty} \mathcal{N}\left(z; \zeta, 1/t\right) \pi(\zeta) d\zeta. \tag{8}$$

*where $\pi$ is a continuous and proper prior density on $\mathbb{R}$, strictly positive in a neighbourhood of $\mu/\sigma$. Then*

$$\mathcal{C}_{\alpha,t}^{\mathtt{BA}}(\overline{Y}_t, \widehat{\sigma}_t; \pi) := \left[\overline{Y}_t \pm \frac{\widehat{\sigma}_t}{\sqrt{t}} \sqrt{\log\left(\frac{t}{2\pi\alpha^2 \eta_t(\overline{Y}_t/\widehat{\sigma}_t)^2}\right)}\right] \tag{9}$$

*forms a $(1-\alpha)$–AsympCS with approximation rate $1/\sqrt{t \log t}$ for $\mu$.*

In Theorem 3, the density $\pi$ encodes prior beliefs about the ratio $\mu/\sigma$. Under this prior, $\eta_t$ represents the marginal density of the standardised mean $\overline{Y}_t/\sigma$ that would arise if the observations $(Y_t)_{t \geq 1}$ were normally distributed. In contrast to the non-assisted AsympCS (6), the width of the Bayes-assisted AsympCS (9) varies with $\overline{Y}_t/\widehat{\sigma}_t$: when the data align with the prior, $\eta_t(\overline{Y}_t/\widehat{\sigma}_t)$ is large and the interval narrows; when they conflict, $\eta_t(\overline{Y}_t/\widehat{\sigma}_t)$ is small and the interval widens. It is worth emphasising that, even when the prior is strongly misspecified, the Bayes-assisted AsympCS (9) remains valid. In the case of a Gaussian prior $\pi$ centred at $\mu_0$ with variance $\tau^2$, we obtain the following AsympCS :

$$\mathcal{C}_{\alpha,t}^{\mathtt{BA}}(\overline{Y}_t, \widehat{\sigma}_t; \mathcal{N}(\cdot; \mu_0, \tau^2)) = \left[\overline{Y}_t \pm \frac{\widehat{\sigma}_t}{\sqrt{t}} \sqrt{\log\left(\frac{t\tau^2 + 1}{\alpha^2}\right) + \frac{(\overline{Y}_t/\widehat{\sigma}_t - \mu_0)^2}{\tau^2 + 1/t}}\right]. \tag{10}$$

Setting $\rho = \tau$ allows a direct comparison between (10) and its non-assisted counterpart (6). When the data agree with the prior – i.e., $\overline{Y}_t/\widehat{\sigma}_t - \mu_0 \simeq 0$ – the Bayes-assisted interval is narrower than the non-assisted one. Conversely, if the data conflict with the prior, $(\overline{Y}_t/\widehat{\sigma}_t - \mu_0)^2$ is large and the Bayes-assisted AsympCS becomes wider than (6). The proof of Theorem 3 is similar to that of [2, Theorem 2.2]. First, note that $\mathcal{C}_{\alpha,t}^{\mathtt{BA}}(\overline{Y}_t, \mathrm{var}(Y); \pi)$ would be an exact CS if the observations were normally distributed. This follows from an application of the method of mixtures for nonnegative martingales, using the prior $\pi$ as mixing density, together with Ville's inequality. Second, we use KMT strong coupling to approximate in an almost sure sense $\overline{Y}_t$ by a sample average of i.i.d. Gaussian random variables. As in the non-assisted case, Theorem 3 can be extended to the non-i.i.d. setting, as long as one can find such a strong coupling.

**Theorem 4** (Asymptotic Bayes-assisted CS – non-i.i.d. case). *Consider the same notation and assumptions as in Theorem 2. Let $\pi$ be a continuous and proper prior density on $\mathbb{R}$, strictly positive in a neighbourhood of $\mu/\sigma$, and let $\eta_t$ be the density (8) for any $t \geq 1$. Then, the sequence of intervals $(\mathcal{C}_{\alpha,t}^{\mathtt{BA}}(\widehat{\mu}_t, \widehat{\sigma}_t; \pi))_{t \geq 1}$ forms a $(1-\alpha)$–AsympCS with approximation rate $1/\sqrt{t \log t}$ for $\mu$.*

## 3.3 Asymptotic Type-I error control

The asymptotic confidence sequences defined above satisfy an asymptotic version of time-uniform Type-I error control (in the sense of [2, §2.5]; see also [24]).

**Theorem 5** (Asymptotic Type-I error control). *Assume the hypotheses of one of Theorems 1 to 4, and let $(\mathcal{C}_{\alpha,t})$ be the corresponding $(1 - \alpha)$–AsympCS for $\mu$. Then*

$$\liminf_{m \to \infty} \Pr\left(\mu \in \mathcal{C}_{\alpha,t} \text{ for all } t \geq m\right) \geq 1 - \alpha. \tag{11}$$

## 4 Control variates and PPI: background and strong coupling

Prediction-powered inference (PPI) closely relates to control variates, a standard variance-reduction method in Monte Carlo estimation [25, §4.1]. In fact, each PPI estimator can be expressed as a control-variate estimator. We begin with a review of control variates and derive a KMT-type strong-coupling result for these estimators, before providing additional background on PPI.

### 4.1 Control variates: definitions and KMT strong coupling

Let $(U, V)$ be real-valued random variables with finite variance, and consider the goal of estimating $\gamma = \mathbb{E}[V]$ from an i.i.d. sample $(U_i, V_i)_{i=1}^n$. If $\mu = \mathbb{E}[U]$ is known, the control-variate estimator (CVE) of $\gamma$ is defined as

$$\widehat{\gamma}_\lambda^{\text{icv}} = \overline{V} - \lambda(\overline{U} - \mu) = \frac{1}{n} \sum_{i=1}^n \left(V_i - \lambda(U_i - \mu)\right), \tag{12}$$

where $\overline{U}$ and $\overline{V}$ denote the empirical means of $(U_i)_{i=1}^n$ and $(V_i)_{i=1}^n$, respectively, $\lambda \in \mathbb{R}$ is a tunable coefficient, and the term $U_i - \mu$ acts as a control variate. The estimator $\widehat{\gamma}_\lambda^{\text{icv}}$ is unbiased, consistent, and has variance $\text{var}(\widehat{\gamma}_\lambda^{\text{icv}}) = (\text{var}(V) - 2\lambda\text{cov}(U, V) + \lambda^2\text{var}(U))/n$. Compared to the standard sample mean estimator $\overline{V}$, which attains variance $\text{var}(\overline{V}) = \text{var}(V)/n$, using $\widehat{\gamma}_\lambda^{\text{icv}}$ yields variance reduction when $\lambda < 2\text{cov}(U, V)/\text{var}(U)$. The minimum variance is achieved at the optimal coefficient $\lambda^\star = \text{cov}(U, V)/\text{var}(U)$, for which $\text{var}(\widehat{\gamma}_{\lambda^\star}^{\text{icv}}) = (1 - \rho_{U,V}^2)\text{var}(\overline{V})$, where $\rho_{U,V}$ is the correlation between $U$ and $V$. That is, stronger correlation leads to greater variance reduction.

In practice, $\mu$ and $\lambda^\star$ are typically unknown. When this is the case, given an additional i.i.d. sample $(\widetilde{U}_j)_{j=1}^{N_n}$, independent of $(U_i, V_i)_{i=1}^n$, where $\widetilde{U}_1$ has the same distribution as $U$, one can estimate $\mu$ by $\widehat{\mu} = \frac{1}{N_n} \sum_{j=1}^{N_n} \widetilde{U}_j$ and plug it into Equation (12). For fixed $\lambda$, this gives

$$\widehat{\gamma}_\lambda^{\text{cv}} = \overline{V} - \lambda(\overline{U} - \widehat{\mu}) = \frac{1}{n} \sum_{i=1}^n \left(V_i - \lambda(U_i - \widehat{\mu})\right). \tag{13}$$

Similarly, $\lambda^\star$ may be estimated from data as $\widehat{\lambda} = \widehat{\text{cov}}((U_i, V_i)_{i=1}^n)/\widehat{\text{var}}((U_i)_{i=1}^n)$, where $\widehat{\text{var}}(\cdot)$ and $\widehat{\text{cov}}(\cdot)$ denote the sample variance and covariance, respectively. Plugging $\widehat{\lambda}$ into (13) defines $\widehat{\gamma}^{\text{cv+}} := \widehat{\gamma}_{\widehat{\lambda}}^{\text{cv}}$, which is similar to the semi-supervised least squares estimator of Zhang et al. [11, Eq. (2.15)]. As discussed in Section 3, deriving an AsympCS requires a strong coupling between the estimator and a sequence of i.i.d. Gaussian random variables. We now establish this coupling, a key ingredient for constructing AsympCS for CVEs (and, in particular, for PPI estimators).

**Proposition 1** (Asymptotics for CVEs). *Assume $\mathbb{E}|U|^{2+\delta}$ and $\mathbb{E}|V|^{2+\delta} < \infty$ for some $0 < \delta < 1$. Then, almost surely as $n \to \infty$,*

$$\widehat{\gamma}^{\text{cv+}} = \widehat{\gamma}_{\lambda^\star}^{\text{cv}} + o\left(\frac{1}{\sqrt{n \log n}}\right) = \overline{V} - \lambda^\star(\overline{U} - \widehat{\mu}) + o\left(\frac{1}{\sqrt{n \log n}}\right). \tag{14}$$

**Proposition 2** (KMT coupling for CVEs). *Assume $\mathbb{E}|U|^{2+\delta}$ and $\mathbb{E}|V|^{2+\delta} < \infty$ for some $0 < \delta < 1$. Additionally, assume that $\left|\frac{n}{N_n} - r\right| = O(1/n^{1-a})$ with $0 < a < 2/(2+\delta)$, for some $r \in [0, 1]$. Then, there exist i.i.d. Gaussian random variables $(W_i^{\text{cv}})_{i \geq 1}$ with mean $\gamma$ and variance*

$$\nu_\lambda^{\text{cv}} := \text{var}(V - \lambda U) + r\text{var}(\lambda U) = \text{var}(V) - 2\lambda\text{cov}(U, V) + \lambda^2(1 + r)\text{var}(U)$$

*such that, almost surely as $n \to \infty$,*

$$\widehat{\gamma}_\lambda^{\text{cv}} = \frac{1}{n} \sum_{i=1}^n W_i^{\text{cv}} + o\left(\frac{1}{\sqrt{n \log n}}\right). \tag{15}$$

*Likewise, there exist i.i.d. Gaussian random variables $(W_i^{\text{cv+}})_{i\geq 1}$ with mean $\gamma$ and variance $\nu^{\text{cv+}} := \nu^{\text{cv}}_{\lambda^\star} = \text{var}(V)\left[1 - (1-r)\rho_{U,V}^2\right]$ such that, almost surely as $n \to \infty$,*

$$\widehat{\gamma}^{\text{cv+}} = \frac{1}{n}\sum_{i=1}^{n} W_i^{\text{cv+}} + o\left(\frac{1}{\sqrt{n\log n}}\right). \tag{16}$$

*The estimators*

$$\widehat{\nu}^{\text{cv}}_\lambda((U_i, V_i)_{i=1}^{n}, (\widetilde{U}_j)_{j=1}^{N_n}) = \frac{1}{n-2}\sum_{i=1}^{n}(V_i - \overline{V} - \lambda(U_i - \overline{U}))^2 + \frac{n\lambda^2}{N_n(N_n-1)}\sum_{j=1}^{N_n}(\widetilde{U}_j - \widehat{\mu})^2 \tag{17}$$

$$\widehat{\nu}^{\text{cv+}}((U_i, V_i)_{i=1}^{n}) = \frac{1 - n/N_n}{n-2}\sum_{i=1}^{n}(V_i - \overline{V} - \widehat{\lambda}(U_i - \overline{U}))^2 + \frac{n/N_n}{n-1}\sum_{i=1}^{n}(V_i - \overline{V})^2 \tag{18}$$

*are consistent estimators of $\nu^{\text{cv}}_\lambda$ and $\nu^{\text{cv+}}$, respectively, where $\widehat{\mu} = \frac{1}{N_n}\sum_{j=1}^{N_n}\widetilde{U}_j$.*

## 4.2 PPI estimators: definitions and asymptotic properties

Owing to Equation (2), the PPI estimator $\widehat{\theta}_n$ is the value of $\theta$ that solves the equation $\widehat{g}_{\theta,n} = 0$, where $\widehat{g}_{\theta,n} = \widehat{m}_{\theta,n} + \widehat{\Delta}_{\theta,n}$ is an estimator of $g_\theta$. Here, $\widehat{m}_{\theta,n}$ and $\widehat{\Delta}_{\theta,n}$ are estimators of $m_\theta$ and $\Delta_\theta$, respectively. A typical choice for $\widehat{m}_{\theta,n}$ is the sample mean of the unlabelled data,

$$\widehat{m}_{\theta,n} = \frac{1}{N_n}\sum_{j=1}^{N_n}\ell'_\theta(\widetilde{X}_j, f(\widetilde{X}_j)). \tag{19}$$

Different choices for $\widehat{\Delta}_{\theta,n}$ have been proposed in the literature, leading to different PPI estimators.

**Standard PPI.** Angelopoulos et al. [1] use the sample mean

$$\widehat{\Delta}^{\text{PP}}_{\theta,n} = \frac{1}{n}\sum_{i=1}^{n}\left(\ell'_\theta(X_i, Y_i) - \ell'_\theta(X_i, f(X_i))\right) \tag{20}$$

as an estimator for $\Delta_\theta$. Combining Equation (20) and Equation (19),

$$\widehat{g}^{\text{PP}}_{\theta,n} = \widehat{m}_{\theta,n} + \widehat{\Delta}^{\text{PP}}_{\theta,n} = \left[\frac{1}{n}\sum_{i=1}^{n}\ell'_\theta(X_i, Y_i)\right] - \left(\left[\frac{1}{n}\sum_{i=1}^{n}\ell'_\theta(X_i, f(X_i))\right] - \widehat{m}_{\theta,n}\right) \tag{21}$$

is a CVE with control variate $\ell'_\theta(X_i, f(X_i)) - \widehat{m}_{\theta,n}$ and control-variate parameter $\lambda = 1$. For the squared loss, the estimator $\widehat{\theta}^{\text{PP}}_n$ solving $\widehat{g}^{\text{PP}}_{\theta,n} = 0$ also takes the control-variate form

$$\widehat{\theta}^{\text{PP}}_n = \frac{1}{n}\sum_{i=1}^{n}Y_i - \left(\frac{1}{n}\sum_{i=1}^{n}f(X_i) - \frac{1}{N_n}\sum_{j=1}^{N_n}f(\widetilde{X}_j)\right), \tag{22}$$

with control variate $f(X_i) - \frac{1}{N_n}\sum_{j=1}^{N_n}f(\widetilde{X}_j)$ and $\lambda = 1$.

**PPI++.** Angelopoulos et al. [7] extend the standard PPI estimator (21) by allowing the control-variate parameter $\lambda$, which they call *power-tuning* parameter, to take values other than 1. The resulting estimator is

$$\widehat{\Delta}^{\text{PP+}}_{\theta,n} = \widehat{\Delta}^{\text{PP}}_{\theta,n} - (\widehat{\lambda}_{\theta,n} - 1)\left(\frac{1}{n}\left[\sum_{i=1}^{n}\ell'_\theta(X_i, f(X_i))\right] - \widehat{m}_{\theta,n}\right), \tag{23}$$

where $\widehat{\lambda}_{\theta,n}$ is the estimator $\widehat{\lambda}_{\theta,n} = \widehat{\text{cov}}\left((\ell'_\theta(X_i, Y_i), \ell'_\theta(X_i, f(X_i)))_{i=1}^{n}\right)/\widehat{\text{var}}\left((\ell'_\theta(X_i, f(X_i)))_{i=1}^{n}\right)$. In this case, $\widehat{\Delta}^{\text{PP+}}_{\theta,n}$ is a CVE with centred control variate $\ell'_\theta(X_i, f(X_i)) - \widehat{m}_{\theta,n}$, which depends only on the black-box predictions. As a result,

$$\widehat{g}^{\text{PP+}}_{\theta,n} = \widehat{m}_{\theta,n} + \widehat{\Delta}^{\text{PP+}}_{\theta,n} = \left[\frac{1}{n}\sum_{i=1}^{n}\ell'_\theta(X_i, Y_i)\right] - \widehat{\lambda}_{\theta,n}\left(\left[\frac{1}{n}\sum_{i=1}^{n}\ell'_\theta(X_i, f(X_i))\right] - \widehat{m}_{\theta,n}\right) \tag{24}$$

is also a CVE. Under the squared loss, we obtain

$$\widehat{\theta}_n^{\text{PP+}} = \frac{1}{n} \sum_{i=1}^n Y_i - \widehat{\lambda}_{0,n} \left( \frac{1}{n} \sum_{i=1}^n f(X_i) - \frac{1}{N_n} \sum_{j=1}^{N_n} f(\widetilde{X}_j) \right), \tag{25}$$

where in this case $\widehat{\lambda}_{\theta,n} = \widehat{\lambda}_{0,n}$ for all $\theta$. Standard asymptotic confidence intervals for PPI and PPI++ rely on CLTs for the estimators $\widehat{g}_{\theta,n}$, $\widehat{m}_{\theta,n}$, and $\widehat{\Delta}_{\theta,n}$. In contrast, constructing asymptotic confidence sequences requires almost sure approximations by averages of i.i.d. Gaussian variables. Since the estimators for $g_\theta$, $m_\theta$ and $\Delta_\theta$ are all CVEs, the asymptotic results of Proposition 1 and the KMT coupling of Proposition 2 both apply.

## 5 Anytime-valid, Bayes-assisted, prediction-powered inference

In this section we combine the results of Sections 3 and 4 within the PPI framework to obtain AsympCS for $g_\theta$. For any $\theta \in \mathbb{R}$ and $i \geq 1$, let $U_{\theta,i} = \ell'_\theta(X_i, f(X_i))$, $\widetilde{U}_{\theta,i} = \ell'_\theta(\widetilde{X}_i, f(\widetilde{X}_i))$ and $V_{\theta,i} = \ell'_\theta(X_i, Y_i)$. Define $\overline{V}_{\theta,n} = \frac{1}{n} \sum_{i=1}^n V_{\theta,i}$ and $\overline{U}_{\theta,n} = \frac{1}{n} \sum_{i=1}^n U_{\theta,i}$. In the following, we assume $\mathbb{E}|U_{\theta,i}|^{2+\delta}$, $\mathbb{E}|\widetilde{U}_{\theta,i}|^{2+\delta}$ and $\mathbb{E}|V_{\theta,i}|^{2+\delta} < \infty$ for some $0 < \delta < 1$, and that $\left| \frac{n}{N_n} - r \right| = O(1/n^{1-a})$ with $0 < a < 2/(2+\delta)$ for some $r \in [0,1]$.

### 5.1 Anytime-valid PPI

We first derive AsympCS that do not incorporate prior information about the black-box predictor's accuracy. The following result follows directly from Proposition 2 and Theorem 2, owing to the control-variate form of the PPI estimator $\widehat{g}_{\theta,n}^{\text{PP}}$ (21) and of the PPI++ estimator $\widehat{g}_{\theta,n}^{\text{PP+}}$ (24).

**Proposition 3.** *Let $\widehat{g}_{\theta,n}$ be either the PPI (21) or the PPI++ (24) estimator. For PPI, let $(\widehat{\sigma}_{\theta,n}^g)^2 = \widehat{\nu}_1^{\text{cv}}((U_{\theta,i}, V_{\theta,i})_{i=1}^n, (\widetilde{U}_{\theta,j})_{j=1}^{N_n})$ (see (17)). For PPI++, let $(\widehat{\sigma}_{\theta,n}^g)^2 = \widehat{\nu}^{\text{cv+}}((U_{\theta,i}, V_{\theta,i})_{i=1}^n)$ (see (18)). Then, for any $\rho > 0$, the sequence of intervals defined as $\mathcal{C}_{\alpha,\theta,n}^g = \mathcal{C}_{\alpha,n}^{\text{NA}}(\widehat{g}_{\theta,n}, \widehat{\sigma}_{\theta,n}^g; \rho)$ forms a $(1-\alpha)$–AsympCS with approximation rate $1/\sqrt{n \log n}$ for $g_\theta$ and asymptotic Type-I error control.*

### 5.2 Anytime-valid, Bayes-assisted, PPI

In many modern applications extremely accurate black-box predictors are available (e.g., [26, 27, 28]). When this is the case, we can leverage this prior information to obtain tighter AsympCS for $g_\theta$ via a zero-mean prior on $\Delta_\theta$. Following the decomposition in Equation (3), we combine an AsympCS for $m_\theta$ (Proposition 4) with a Bayes-assisted AsympCS for $\Delta_\theta$ (Proposition 5).

**Proposition 4** (AsympCS for $m_\theta$). *Let $\widehat{m}_{\theta,n}$ and $(\widehat{\sigma}_{\theta,n}^f)^2$ be the sample mean (19) and sample variance of $(\ell'_\theta(\widetilde{X}_j, f(\widetilde{X}_j)))_{j=1}^{N_n}$. Let $\delta \in (0,1)$. For any $\rho > 0$, $\mathcal{R}_{\delta,\theta,n} = \mathcal{C}_{\delta,n}^{\text{NA}}(\widehat{m}_{\theta,n}, \widehat{\sigma}_{\theta,n}^f; \rho)$ forms a $(1-\delta)$–AsympCS with approximation rate $1/\sqrt{n \log n}$ for $m_\theta$ and asymptotic Type-I error control.*

**Proposition 5** (Bayes-assisted AsympCS for $\Delta_\theta$). *For PPI, let $\widehat{\Delta}_{\theta,n}$ and $(\widehat{\sigma}_{\theta,n}^\Delta)^2$ be the sample mean (20) and sample variance of $(V_{\theta,i} - U_{\theta,i})_{i=1}^n$. For PPI++, let $\widehat{\Delta}_{\theta,n}$ be the control-variate estimator (23) and $(\widehat{\sigma}_{\theta,n}^\Delta)^2 = \widehat{\nu}^{\text{cv+}}((U_{\theta,i}, V_{\theta,i} - U_{\theta,i})_{i=1}^n)$ (see (18)). Let $\kappa \in (0,1)$. For any continuous proper prior $\pi$, the sequence of Bayes-assisted intervals $\mathcal{T}_{\kappa,\theta,n} = \mathcal{C}_{\kappa,n}^{\text{BA}}(\widehat{\Delta}_{\theta,n}, \widehat{\sigma}_{\theta,n}^\Delta; \pi)$ forms a $(1-\kappa)$–AsympCS with approximation rate $1/\sqrt{n \log n}$ for $\Delta_\theta$ and asymptotic Type-I error control.*

Finally, for both PPI and PPI++, the confidence sequences $\mathcal{R}_{\delta,\theta,n}$ and $\mathcal{T}_{\alpha-\delta,\theta,n}$ are combined via a Minkowski sum to obtain a $(1-\alpha)$–AsympCS for $g_\theta$, with approximation rate $1/\sqrt{n \log n}$ and asymptotic Type-I error control, of the form

$$\mathcal{C}_{\alpha,\theta,n}^g = \left[ \widehat{g}_{\theta,n} \pm \left\{ \frac{\widehat{\sigma}_{\theta,n}^\Delta}{\sqrt{n}} \sqrt{\log \left( \frac{n(2\pi\kappa^2)^{-1}}{\eta_n (\widehat{\Delta}_{\theta,n}/\widehat{\sigma}_{\theta,n}^\Delta)^2} \right)} + \frac{\widehat{\sigma}_{\theta,n}^f}{\sqrt{N_n}} \sqrt{\frac{1 + N_n\rho^2}{N_n\rho^2} \log \left( \frac{N_n\rho^2 + 1}{\delta^2} \right)} \right\} \right] \tag{26}$$

where $\widehat{g}_{\theta,n}$ is either the PPI estimator (21) or the PPI++ estimator (24). Solving Equation (4) gives the confidence region for $\theta^\star$. In the case of the squared loss, $\mathcal{C}_{\alpha,n}^{\mathrm{avpp}}$ is an interval, given by

$$\mathcal{C}_{\alpha,n}^{\mathrm{avpp}} = \left[\widehat{\theta}_n \pm \left\{ \frac{\widehat{\sigma}_{0,n}^{\Delta}}{\sqrt{n}} \sqrt{\log\left(\frac{n}{2\pi\kappa^2 \eta_n (\widehat{\Delta}_{0,n}/\widehat{\sigma}_{0,n}^{\Delta})^2}\right)} + \frac{\widehat{\sigma}_{0,n}^{f}}{\sqrt{N_n}} \sqrt{\frac{1+N_n\rho^2}{N_n\rho^2} \log\left(\frac{N_n\rho^2 + 1}{\delta^2}\right)} \right\} \right]$$

(27)

where $\widehat{\theta}_n$ is either the PPI estimator (22) or the PPI++ estimator (25).

## 6 Experiments

We compare the PPI and PPI++ AsympCS procedures introduced in Section 5 – with and without Bayes assistance – to the AsympCS relying solely on labelled data (obtained from Theorem 1 and referred to as "classical") on several estimation problems. Bayes-assisted methods are annotated with (G), (L), or (T) to indicate Gaussian, Laplace, or Student-t priors with mean zero and scale depending on the task and reported in the Supplementary Material. For the Student-t prior, we set the degrees of freedom to 2 in all experiments. Since PPI is motivated by settings with scarce labelled data and abundant unlabelled data, we consider the following experimental setting: labelled data arrive sequentially, i.e., $n = 1, 2, \ldots$, while a large unlabelled dataset is available from the start, i.e., $N_n = N$ for all $n$, with $N \gg n$ large enough to exclude any uncertainty on the measure of fit $m_\theta$. As discussed by Cortinovis and Caron [6], this simplifies the comparison between non-assisted and Bayes-assisted PPI, as it rules out any potential loss of efficiency due to the Minkowski sum (26), thereby isolating the effect of the Bayes correction on the CS procedure. For synthetic data, we set $N = \infty$ to guarantee the simplification holds. For real data, we empirically verify that $N$ is large enough to justify this assumption by confirming that anytime validity is preserved – specifically, that the cumulative miscoverage rate remains below the chosen threshold $\alpha = 0.1$ for all $n$. As with CLT-based CIs, the $n$ at which one starts counting the cumulative miscoverage rate of an asymptotic CS is inherently arbitrary; unless otherwise stated, we choose $n = 40$, as we empirically find this to be a reasonably small labelled sample size at which the KMT coupling generally provides a good approximation.

### 6.1 Synthetic data

The synthetic experiments follow a general structure: we start with $N = \infty$ unlabelled samples $\{\widetilde{X}_j\}_{j=1}^N \overset{\mathrm{iid}}{\sim} \mathbb{P}_X$ and successively sample $n$ labelled observations $(X_i, Y_i)_{i=1}^n \overset{\mathrm{iid}}{\sim} \mathbb{P}$ with the goal of estimating the mean $\theta^\star = \mathbb{E}[Y]$.

**Noisy predictions.** This experiment demonstrates that our method can adapt to varying correlation levels between predictions and true labels by using the PPI++ estimator (23). We sample $Y_i \overset{\mathrm{iid}}{\sim} \mathcal{N}(0,1)$, so that $\theta^\star = \mathbb{E}[Y] = 0$. The prediction rule is defined as $f(X_i) = Y_i + \epsilon_i$, where $X_i$ is only used for indexing and $\epsilon_i \overset{\mathrm{iid}}{\sim} \mathcal{N}(0, \sigma_Y^2)$, with the noise level $\sigma_Y \in \{0.1, 0.8, 3\}$. In this case, the optimal control-variate parameter is given by $\lambda_\theta^\star = \lambda^\star = \mathrm{cov}(Y, f(X))/\mathrm{var}(f(X)) = (1 + \sigma_Y^2)^{-1}$, which decreases with $\sigma_Y$. Figure 1 compares the interval volume achieved by classical and non-assisted CS procedures as a function of $n$, while results under informative priors are reported in Section S7.1. For small noise levels, PPI and PPI++ achieve similar performance, and greatly outperform classical inference. As the noise level grows, the machine learning predictions become less informative and standard PPI loses ground to the classical CS. By contrast, PPI++ adapts to the noise level and always performs similarly to, or better than, the other baselines.

**Biased predictions.** This experiment illustrates the potential benefits of incorporating prior information into our method. We sample $X_i \overset{\mathrm{iid}}{\sim} \mathcal{N}(0,1)$ and $Y_i = X_i + \epsilon_i$, where $\epsilon_i \overset{\mathrm{iid}}{\sim} t_{\mathrm{df}}(0,1)$, so that $\theta^\star = \mathbb{E}[Y] = 0$. The prediction rule is defined as $f(X_i) = X_i + \upsilon$, where $\upsilon \in \mathbb{R}$ controls its bias level. For all $\upsilon$, $\lambda^\star = 1$, so PPI and PPI++ coincide. We vary $\upsilon$ between $-1.2$ and $1.2$, and $\mathrm{df} \in \{5, 10, \infty\}$ to study the impact of bias level and noise distribution on the AsympCS procedures. Figure 2 compares the average interval volumes at $n = 100$ as a function of $\upsilon$ for each value of df. Classical inference and non-assisted PPI volumes remain essentially constant across bias levels, reflecting their lack of prior information, and with the latter consistently outperforming the former

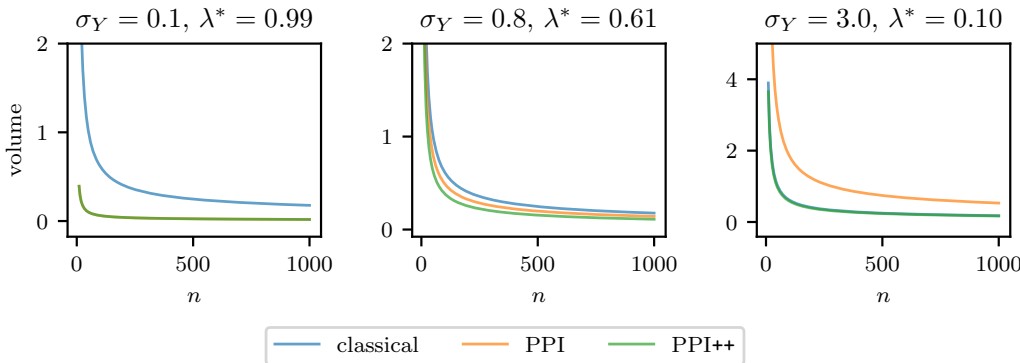

Figure 1: Noisy predictions study. The left, middle and right panels show average interval volume over 1000 repetitions as a function of the labelled sample size $n$ for noise levels $\sigma_Y \in \{0.1, 0.8, 3.0\}$.

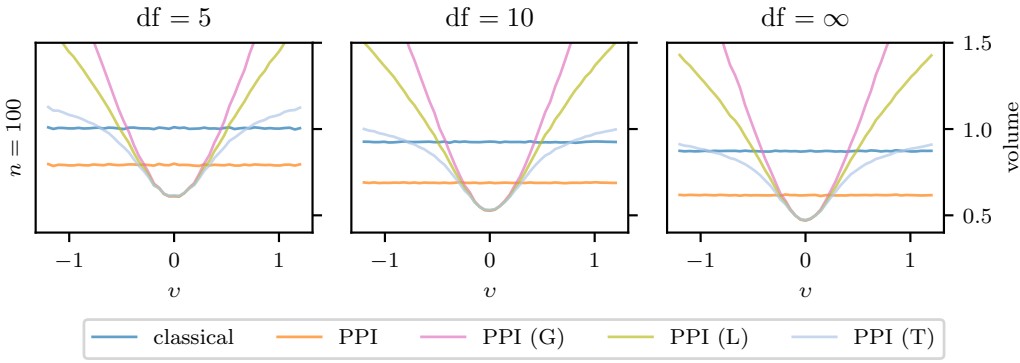

Figure 2: Biased predictions study. The left, middle and right panels show average interval volume over 100 repetitions as a function of the bias level $v$ for df $= 5, 10, \infty$.

by leveraging imputed predictions. On the other hand, the volume of the Bayes-assisted procedures varies widely with the bias level $v$: it is reduced for small $v$, but grows with $|v|$ as the priors become increasingly misspecified. Notably, the volume under the Gaussian prior inflates the fastest with $|v|$, while heavier-tailed Laplace and Student-t priors offer comparatively greater robustness. These conclusions hold for all values of df, which controls the accuracy of the KMT coupling approximation for a given $n$. Coverage results in Section S7.1 show that, while smaller values of df lead to slightly worse coverage, the approximation quality is overall satisfactory in this example.

## 6.2   Real data

We evaluate our method on several real-world datasets, which are described in Section S6.2. While each dataset is, in principle, static (providing label/prediction pairs $(Y_i, f(X_i))_{i=1}^{N+n_1}$), we simulate an online setting akin to Section 6.1 by randomly splitting the data into a labelled set of size $n_1$, serving as a labelled data stream, and an unlabelled set of size $N$.

Figure 3 compares classical and PPI++ AsympCS procedures on the FLIGHTS, FOREST, and GALAX-IES datasets, where the goal is mean estimation. By taking advantage of the unlabelled data, PPI methods consistently yield smaller regions than the classical counterpart, while maintaining reliable coverage. Moreover, Bayes-assisted approaches further improve efficiency for moderate labelled sample sizes, as the quality of the predictions is generally high in these datasets.

Figure S8 reports results for three additional estimation tasks: linear regression (CENSUS), logistic regression (HEALTHCARE), and quantile estimation (GENES). For the first two tasks, the same conclusions as for mean estimation hold: PPI methods consistently outperform classical inference,

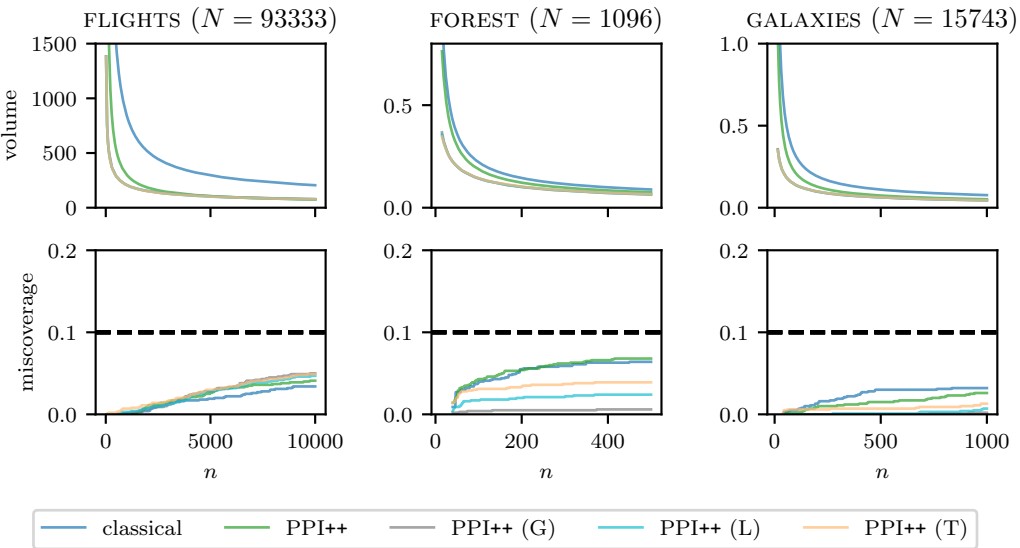

Figure 3: Mean estimation. The top and bottom rows show the average interval volume and cumulative miscoverage rate over 1000 repetitions for the FLIGHTS, FOREST, and GALAXIES datasets.

with Bayes-assisted approaches providing an additional efficiency boost. For the quantile estimation task, non-assisted PPI still improves over classical inference by leveraging the machine learning predictions; however, the Bayes-assisted methods yield larger regions than the other approaches, reflecting lower prediction quality in this dataset.

## 7 Discussion

We extended the PPI framework to the sequential setting via asymptotic confidence sequences, which allow for the seamless integration of prior information about the quality of the auxiliary predictions. However, several directions merit further investigation. The results developed here are for scalar parameter values $\theta$. Extensions to multivariate settings are discussed in Section S4, building on earlier work by Waudby-Smith et al. [2, §B.10]. In the non-assisted case, we focused on asymptotic confidence sequences of the form (6), but other options are possible. In particular, as discussed in Section S8, the parameter-free CS proposed by Wang and Ramdas [29], which is based on an improper prior, may be used as an exact reference CS in place of Equation (6).

The AsympCS derived in this paper are asymptotically valid for i.i.d. data under mild, nonparametric assumptions. Promising directions include extensions to non-i.i.d. observations, as well as the development of *nonasymptotic*, nonparametric Bayes-assisted confidence sequences under stricter assumptions (e.g., bounded means), building on the work of Waudby-Smith and Ramdas [15]. In the non-assisted case, the parameter $\rho$ was assumed to be fixed. Waudby-Smith et al. [2, §2.5] considered delayed-start sequences $\mathcal{C}_{\alpha,t}(m)$ that may depend on the start time $m$; this includes allowing the tuning parameter $\rho$ to depend on $m$. Their asymptotic Type-I error control result, derived under assumptions similar to those used here, also applies in our setting. Another interesting direction would be to adapt similar ideas to the Bayes-assisted construction.

PPI AsympCS procedures share the computational considerations of their fixed-time counterparts. Beyond mean estimation (e.g., Figure S8), they typically require constructing a grid over $\theta$. When the marginal density $\eta_t$ is not available in closed form (e.g., for the Student-$t$ prior), the Bayes-assisted version requires numerical integration. If computation is a concern, the Laplace prior offers a good compromise: it has heavier tails than the Gaussian while still admitting a closed-form expression for $\eta_t$.

## Acknowledgments and Disclosure of Funding

Valentin Kilian is supported by the Clarendon Funds Scholarship. Stefano Cortinovis is supported by the EPSRC Centre for Doctoral Training in Modern Statistics and Statistical Machine Learning (EP/S023151/1). The authors thank the reviewers for their time and valuable feedback, especially the suggestion to incorporate a discussion on Type-I error control.

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
