# OpenReview forum: "Anytime-valid, Bayes-assisted, Prediction-Powered Inference"
_NeurIPS.cc/2025/Conference — NeurIPS 2025 poster_

### Official Review · Reviewer_BzQ5 · 2025-06-26

**Clarity:** 2
**Significance:** 3
**Originality:** 3
**Rating:** 5
**Confidence:** 2

**Summary:**

This paper introduces a framework for performing statistically valid inference based on machine learning prediction and valid at any point in time. This is part of the Prediction-Powered Inference (PPI) framework, where the general idea is to leverage large unlabeled dataset to improve stastistical efficiency. The main limitation of the existing approach is that it works at "fixed-time".

The main contribution is the development of anytime-valid PP confidence sequences for the sequential setting. Importantly, these sequence keep their statistical coverage guarantees across all sample sizes. This is achieved by connecting the PPI framework to the theory of control variates and with modern tools for constructing asymptotic confidence sequences, which is established in multiple propositions and derivations throughout the paper.

Another contribution is the Bayes-assisted extension of the method. This uses the "method of mixtures" to incorporate Bayesian priors about the accuracy of the prediction. Crucially, when the prior is veridical, confidence intervals can become narrower; but the CI remain valid even if the prior is misspecified.

Finally, the paper validates their method with some experiments on synthetic an real data, showing the efficacy of their approach.

**Questions:**

As mentioned above, the main issue with the paper is its density and it would be unfair to penalize the submission for that, although I do believe that this paper would fare much better in a longer journal format. So I guess this is not very actionable.

What is more actionable is strengthening the empirical validation, for example presenting results beyond the mean estimator, and discussing whether and how these results could be extended in the future to a non-scalar quantity (this is only briefly mentioned in passing in the Discussion).

**Ethical Concerns:**

["NO or VERY MINOR ethics concerns only"]

**Final Justification:**

The work is theoretically sound and interesting, and the authors have provided additional evidence and results that addressed concerns.

**Limitations:**

The authors might highlight more (unless they address it) some of the limitations of the paper, such as the focus on scalar quantities.

**Quality:**

3

**Strengths And Weaknesses:**

This is a strong, extremely dense theoretical paper tackling an interesting problem and combining state-of-the-art results of interest at the intersection of statistics and machine learning. This is a high-quality contribution, significant and original, to my knowledge and understanding -- admittedly quite limited in this field.

The only major issue I found is that the paper is clarity, in that the paper is quite dense and possibly not very well suited for the conference format, whereas it would likely shine more in a journal format with adequate space to breathe. The lack of space is a major constraint and, at least for a non-expert, the reading is quite painful with a non-stop barrage of definitions and propositions, with little explanation or justification. This is not the authors' fault, whose writing is otherwise quite crisp and it's clear they could do better given more space.

The other issue I found is somewhat limited experimental validation. The theory is general, but the empirical validation focuses on scalar mean estimation problems. It seems that extending beyond the mean would require additional approximations (e.g., constructing a grid of values), but these are just mentioned in the Discussion. Also note that another point worth mentioning more clearly is that the entire theory focuses on a *scalar* estimate.

### Minor comments

- Shouldn't the $\eta_n$ term in Eqs (22) and (23) be squared (following from Eq. 8)?

---

> ### Author Rebuttal · Authors · 2025-07-31
>
> We sincerely thank the reviewer for their positive and constructive feedback, and for recognizing our work as a ``high-quality contribution, significant and original." We appreciate the insightful comments, which will help us improve the paper. We address the specific points below.
>
> ### About the technical depth of the paper:
>
> We agree with the reviewer that the paper is dense due to the nature of the topic and the space constraints of the conference format. We appreciate the reviewer's understanding on this front. To improve the paper's accessibility for readers less familiar with this area, we will incorporate additional background material on Prediction-Powered Inference (PPI) and Confidence Sequences (CS) to make the paper more self-contained.
>
> ### Expanding the Empirical Validation
> We thank the reviewer for the suggestion on how to strengthen the impact of our work. Indeed, our method, like standard PPI, can be applied to any non-degenerate convex estimation task. To show this, as well as to illustrate the performance of our method when the predictor provided is less accurate, we have experimented with the following additional tasks using datasets discussed in [1]:
> - HEALTHCARE ($N = 316215$), whose task is to estimate the parameters of a **logistic regression model**, where the outcome is a health insurance indicator and the covariate is income, and for which the predictor is good.
> - CENSUS ($N = 378091$), whose task is to estimate the parameters of a **linear regression model**, where the outcome is income and the covariate is age, and for which the predictor is poor.
> - GENES ($N = 59150$), whose task is to estimate the **median expression level** of a population of genes, and for which the predictor is poor.
>
> The experimental procedure follows the same design as in Section 6.2. The table below compares anytime-valid classical inference, unassisted PPI, and Bayes-assisted PPI in terms of the width of the resulting confidence regions at $n = 500, 1000, 2000$ labelled samples, averaged over $100$ repetitions.
>
> | $n = 500/1000/2000$ | HEALTHCARE ($\times 10^{-5}$) | CENSUS ($\times 10^2$) | GENES ($\times 10^{0}$) |
> |---|---|---|---|
> | classical | 2.3/1.5/1.0 | 4.89/3.31/2.29 | 2.02/1.33/0.92 |
> | PPI | 2.0/1.3/0.8 | 4.09/2.79/1.92 | 1.64/0.99/0.67 |
> | PPI (G) | 1.5/1.1/0.7 | 5.79/5.17/4.28 | 3.18/3.05/2.89 |
> | PPI (L) | 1.5/1.1/0.7 | 5.28/4.17/3.05 | 2.90/2.50/1.68 |
> | PPI (T) | 1.5/1.1/0.8 | 4.90/3.75/2.71 | 2.30/1.57/1.09 |
>
> In particular, unassisted PPI outperforms classical inference on all datasets, showing the potential benefit of PPI in the presence of correlation between predictions and true labels. On the other hand, Bayes-assisted PPI outperforms unassisted PPI on the HEALTHCARE dataset, where the predictor is accurate, but not on the CENSUS and GENES datasets, for which the zero-centered priors are mispecified. In these cases, the behaviour of the three priors considered (Gaussian, Laplace, and Student-t) differs significantly. In particular, as expected, heavier-tailed priors suffer less from the mispecification, achieving a lower width than the Gaussian prior, which is the most sensitive to the predictor's accuracy.
>
> While this is not currently possible for this rebuttal, we will include these results in the form of figures similar to Figure 3 in Section 6.2 to our revised supplementary material. We believe that these results clarify the broad applicability of our method and shed light on the practical implications of prediction accuracy and prior choice.
>
> ### Clarification on the Multivariate Extension
> We thank the reviewer for raising this important point and apologize that our presentation made this seem like a limitation. Our framework is not limited to scalar quantities.
>
> A multivariate generalization of our theory is indeed available, but, given its greater technical depth, and for the sake of presentation clarity, is provided in Section S5 of the supplementary material. We recognize this was not sufficiently highlighted in the main text. In the revision, we will update the main paper, particularly the discussion section, to make the existence and location of these important multivariate results much more prominent.
>
> Additionally, to illustrate the multivariate extension of our method in practice, we study a multivariate version of the mean estimation task with biased predictions in Section 6.1. In particular, we sample $5$-dimensional $X_i \overset{iid}{\sim} \mathcal{MVN}(\mathbf{0}, \Sigma)$, where $\Sigma \in \mathbb{R}^{5 \times 5}$ is a Toeplitz covariance matrix with entries $\Sigma_{ij} = 0.5^{|i-j|}$, and define $Y_i = X_i + \epsilon_i$, where $\epsilon_i$ is sampled i.i.d.~from a standard normal distribution, so that $\theta^* = \mathbb{E}[Y] = \mathbf{0}$. Then, we proceed as in Section 6.1 and define biased predictions $f(X_i) = X_i + \nu$, where $\nu \in \mathbb{R}$ controls the bias level of the predictor.
>
> The table below compares anytime-valid multivariate classical inference, unassisted PPI, and Bayes-assisted PPI under a Gaussian prior in terms of the volume of the resulting confidence ball at $n = 100$ and $N = \infty$ when varying $\nu$ between $0$ and $5.0$, averaged over $1000$ repetitions.
>
> | $\nu$ | 0.0 | 1.0 | 2.0 | 3.0 | 4.0 | 5.0 |
> |---|---|---|---|---|---|---|
> | classical | 252.6 | 256.4 | 251.3 | 254.3 | 252.7 | 252.3 |
> | PPI | 24.4 | 24.2	| 24.4 | 24.0 | 24.3 | 24.1 |
> | PPI (G) | 0.06 | 0.36 | 3.8 | 21.6 | 85.9 | 241.4 |
>
> The behaviour of the methods considered follows the same pattern as in the scalar case. In particular, unassisted PPI always outperforms classical inference, while maintaining roughly the same volume across all values of $\nu$. On the other hand, Bayes-assisted PPI outperforms unassisted PPI when the predictor is accurate ($\nu = 0,\dots,3$), but eventually suffers from the mispecification of the zero-centered prior when the predictor is poor ($\nu = 4,5$).
>
> While this is not currently possible for the rebuttals, we will include this result in the form of a figure similar to Figure 2 in Section 6.2 to our revised supplementary material.
>
> ### Correction of Typo in Equations
>
> The reviewer is absolutely correct. The term $\eta_t$ in Equations (22) and (23) should indeed be squared, as it follows from the variance definition in Equation (8). We are very grateful for this careful catch and will correct the typo in the revision.
>
> We believe these changes will address the reviewer's concerns and improve the paper's clarity and impact. We thank the reviewer again for their valuable guidance.
>
> [1] Anastasios N. Angelopoulos and Stephen Bates and Clara Fannjiang and Michael I. Jordan and Tijana Zrnic. Prediction-Powered Inference. arXiv:2301.09633, 2023.

---

> > ### Comment · Reviewer_BzQ5 · 2025-08-01
> > **Response to reviewer**
> >
> > I thank the authors for the detailed and almost overwhelming rebuttal, including clarifications and additional results. I am confident that if accepted they will make good use of the additional page to improve the clarity of presentation. I am happy to recommend acceptance for the paper.

---

### Official Review · Reviewer_1TNz · 2025-07-02

**Clarity:** 4
**Significance:** 3
**Originality:** 2
**Rating:** 5
**Confidence:** 4

**Summary:**

The authors propose asymptotic anytime-valid confidence sequences that leverage (i) user-specified priors for the signal-to-noise ratio (defined by mean / standard deviation) and (ii) black-box predictors for variance reduction. The work first provides closed-form asymptotic confidence sequences for a user-specified prior with similar techniques to existing works in the literature. Building on top of these results, the authors apply their prior-based confidence sequences to PPI-based estimation, and provide theoretical results on the validity of their approach. Using both synthetic and real-world examples, the authors demonstrate when their approach outperforms existing methods in the literature, highlighting both the benefits and drawbacks of using the PPI-based framework and user-specified priors.

**Questions:**

* A large benefit of anytime-valid and asymptotic anytime-valid methods are that they remain valid beyond i.i.d. data collection (i.e. data collected under time-varying sampling policies such as contextual bandits). In such settings, do the proposed approaches maintain their anytime-validity guarantees?
* As with traditional mixture-martingale methods, the user-specified prior on the signal-to-noise ratio must be specified in advance by the analyst. Are there potential extensions that update the prior in an online fashion, while still maintaining the desired anytime-validity guarantees?

**Ethical Concerns:**

["NO or VERY MINOR ethics concerns only"]

**Final Justification:**

For the recommended score, the authors provided a detailed response regarding conditions for type I error control that were not present in the original manuscript. These additions significantly strengthen its theoretical contributions, and therefore, I would recommend acceptance.

**Limitations:**

As pointed out in weaknesses, the authors use a definition of asymptotic anytime validity that does not explicitly provide any type I error guarantees - only that the sequence converges to some unknown anytime-valid confidence sequence. Generally, asymptotic type I error guarantees require stronger conditions than the goals of Defn 2. I would be willing to raise my score if the authors could elaborate on the additional assumptions needed to provide error guarantees across their approaches.

**Paper Formatting Concerns:**

I have no paper formatting concerns.

**Quality:**

3

**Strengths And Weaknesses:**

**Strengths**
* The authors provide novel results for constructing asymptotic confidence sequences with user-specified priors and methods for leveraging black-box predictors for variance reduction. In settings where a plethora of data and/or models are available before experimentation, the approaches provided in this work enable reductions in confidence sequence width.
* The guarantees of this approach are valid even under poor-quality/noisy predictors, making them applicable across many settings. In particular, while performance depends on the quality of the black-box predictors, the validity of the confidence sequences do not, allowing them to be a potentially conservative option regardless of setting.

**Weaknesses**
* While the authors demonstrate that their approaches are asymptotic confidence sequences (as defined in Defn. 2), they do not provide any type I error guarantees regarding the performance of their approach. Note that to provide error guarantees (such as anytime-valid type I error protection beyond a burn-in time diverging to infinity), one generally requires stronger assumptions than achieving the goals of Definition 2, as discussed in [1] and Assumption 3-$\eta$ in [2].
* The title may be slightly misleading: this work is only concerned with asymptotic anytime-valid inference, rather than standard anytime-valid methods that protect type I error across the entire experiment horizon.

[1] Aurelien Bibaut, Nathan Kallus, and Michael Lindon. Near-optimal non-parametric sequential tests and confidence sequences with possibly dependent observations, 2024.
[2] I. Waudby-Smith, D. Arbour, R. Sinha, E. H. Kennedy, and A. Ramdas. Time-uniform central limit theory and asymptotic confidence sequences. The Annals of Statistics, 52(6):2613–2640, 2024.

---

> ### Author Rebuttal · Authors · 2025-07-31
>
> We sincerely thank the reviewer for their thoughtful and constructive feedback. Their comments are insightful and will help us improve our manuscript. We address each point below.
>
>
>
> ### On Asymptotic Type I Error Guarantees
>
> We thank the reviewer for highlighting the question of Type I error control. After further investigation, we confirm that one can have type-I error control protection beyond a burn-in time diverging to infinity. As the reviewer anticipated, and as in Bibaut et al. (2024) and Waudby-Smith et al. (2024), this result requires slightly stronger assumptions than those stated in the main text. Fortunately, these stronger conditions already appear in Section S4 of the supplementary material, where they underpin confidence sequences with sharper convergence rates.
>
> The key assumption (see Section S4.4) is to assume that the random variables $U_{\theta,i}=\ell_\theta'(X_i,f(X_i))$ and $V_{\theta,i}=\ell'_\theta(X_i,Y_i)$ possess absolute moments of order $2+\delta$, for some $\delta>0$. Under this condition, we obtain the following theorem for the anytime-valid PPI confidence sequence presented in Section 5.1; a sketch of the proof is provided below. An analogous result holds for the Bayesian procedures of Section 5.2 under the same assumptions, in addition to the already stated assumptions on the prior.  We will incorporate these theorems into Section S4 of the supplement. We hope this addresses the reviewer's concern.
>
> __Theorem__ [Type-1 control for non-assisted PPI confidence sequence]. Assume $\mathbb E[|U_{\theta,i}|^{2+\delta}]<\infty$ and $\mathbb E[|V_{\theta,i}|^{2+\delta}]<\infty$ for some $\delta>0$. Assume that $|n/N_n-r|=O(1/n^{1-a})$ with $0<a<2/(2+\delta)$ for some $r\in[0,1]$. Let $\widehat g_{\theta,n}$ and $\widehat \sigma_{\theta,n}^g$ be as defined in Proposition 3 of our paper. Then, for any $\rho>0$,
>
> $$
> \liminf_{m\to\infty} \Pr(\forall n\geq m~~g_\theta\in \mathcal C_{\alpha,\theta,n}^{\text{NA}}(\widehat g_{\theta,n},\widehat\sigma_{\theta,n}^g,\rho))\geq 1-\alpha.
> $$
>
> Furthermore, for $\rho_m = \sqrt{\frac{-2\log\alpha+\log(-2\log\alpha)+1}{m\log(m\vee e)}}$ choosen as in Section 2.5 of Waudby-Smith et al. (2024), we have:
>
> $$
> \liminf_{m\to\infty} \Pr(\forall n\geq m,~~g_\theta\in \mathcal C_{\alpha,\theta,n}^{\text{NA}}(\widehat g_{\theta,n},\widehat\sigma_{\theta,n}^g,\rho_m))\geq 1-\alpha.
> $$
>
> __Sketch of proof__. By the Marcinkiewicz-Zygmund strong law of large numbers, we have $\widehat \sigma_{\theta,n}^g-\sigma_{\theta}^g=o(1/\log n)$ for some $\sigma_\theta^g$. Additionally, by the results of Section S4 in the supplementary material, the stated assumptions imply that the control-variate estimator $\widehat g_{\theta,n}$ satisfies the tighter strong coupling in Equation (S18)
>
> $$
> \widehat g_{\theta,n}=g_\theta +  \frac{\sigma_{\theta}^g}{n}\sum_{i=1}^n W_i +o(\frac{1}{\sqrt{n\log n}})
> $$
>
> for some i.i.d. zero-mean, unit-variance, Gaussian random variables $(W_i)$. The proof then follows similarly from that in Appendix A7 of Waudby-Smith et al. (2024). Writing $\widehat g_{\theta,t}'$=$\widehat g_{\theta,\lfloor t\rfloor}$ and $\widehat \sigma_{\theta,t}^{g}{}'$=$\widehat \sigma_{\theta,\lfloor t\rfloor}^{g}$ for $t\in\mathbb R,t\geq 1$, we have the continuous-time coupling
>
> $$
> \widehat g_{\theta,t}'=g_\theta +  \frac{\sigma_{\theta}^g}{t}B_t +o(\frac{1}{\sqrt{t\log t}})
> $$
>
> where $B_t$ is a standard Brownian motion. We have
>
> $$
> \Pr(\forall n\geq m,~g_\theta\in \mathcal C_{\alpha,\theta,n}^{\text{NA}}(\widehat g_{\theta,n},\widehat\sigma_{\theta,n}^g,\rho))
> $$
>
> $$
> =\Pr(\forall t\geq \mathbb R^{\geq m},~|\widehat g_{\theta,t}'-g_\theta|\leq \frac{\widehat\sigma_{\theta,t}^g{}'}{\sqrt{t}}\sqrt{(1+\frac{1}{t\rho^2})\log(\frac{t\rho^2+1}{\alpha^2})} )
> $$
>
> $$
> =\Pr(\forall t\in \mathbb R^{\geq m},~|\frac{\sigma_{\theta}^g}{t}B_t +o(\frac{1}{\sqrt{t\log t}})|\leq \frac{\sigma_{\theta}^g}{\sqrt{t}}\sqrt{(1+\frac{1}{t\rho^2})\log(\frac{t\rho^2+1}{\alpha^2})} + o(\frac{1}{\sqrt{t\log t}} ))
> $$
>
> $$
> =\Pr(\forall t\in \mathbb R^{\geq m},~|\frac{\sigma_{\theta}^g}{t}B_t +o(\frac{1}{\sqrt{t\log t}})|\leq \frac{\sigma_{\theta}^g}{\sqrt{t}}\sqrt{(1+\frac{1}{t\rho^2})\log(\frac{t\rho^2+1}{\alpha^2})} + o(\frac{1}{\sqrt{t\log t}} ))
> $$
>
> $$
> =\Pr(\forall t\in \mathbb R^{\geq m},~|B_t +o(\frac{\sqrt{t}}{\sqrt{\log t}})|\leq \sqrt{(t+\frac{1}{\rho^2})\log(\frac{t\rho^2+1}{\alpha^2})} + o(\frac{\sqrt{t}}{\sqrt{\log t}} ))
> $$
>
> Letting $t=s d_m$ where $d_m=m\log m$, the above probability can be written as
>
> $$
> \Pr(\forall s\geq 1/\log m,~|B_s +o(\sqrt{\frac{s}{\log s d_m}})|\leq \sqrt{(s+\frac{1}{d_m \rho^2})\log(\frac{s d_m \rho^2+1}{\alpha^2})} + o(\sqrt{\frac{s}{\log s d_m}}) )
> $$
>
> For a fixed $\rho$, the right-handside bound goes to infinity as $m\to\infty$. Hence the probability goes to 1 and satisfies the type-I coverage. If $\rho=\rho_m$, we obtain
>
> $$
> \Pr(\forall s\geq 1/\log m,~|B_s +o(\sqrt{\frac{s}{\log s d_m}})|\leq \sqrt{(s+\frac{1}{c_\alpha})\log(\frac{s c_\alpha+1}{\alpha^2})} + o(\sqrt{\frac{s}{\log s d_m}}) )
> $$
>
> where $c_\alpha=-2\log\alpha+\log(-2\log\alpha)+1$. Using standard results on Brownian motion (Robbins and Siegmund):
>
> $$
> \Pr(\forall s\geq 0,~|B_s|\leq \sqrt{(s+\frac{1}{c_\alpha})\log(\frac{s c_\alpha+1}{\alpha^2})})=1-\alpha,
> $$
>
> we obtain the required lower bound.
>
>
>
>
>
>
>
> ### On the Title
>
> We agree that it is important to state the *asymptotic* nature of our guarantees explicitly. To make this clear from the outset, we will revise the abstract as follows.
>
> **Revised Abstract:**
>
> >Given a large pool of unlabelled data and a smaller amount of labels, prediction-powered inference (PPI) leverages machine learning predictions to increase the statistical efficiency of standard confidence interval procedures based solely on labelled data, while preserving their fixed-time validity.
> In this paper, we extend the PPI framework to the sequential setting, where labelled and unlabelled datasets grow over time.
> Exploiting Ville's inequality and the method of mixtures, we propose prediction-powered confidence sequence procedures that are **asymptotically** valid uniformly over time and naturally accommodate prior knowledge on the quality of the predictions to further boost efficiency.
> We carefully illustrate the design choices behind our method and demonstrate its effectiveness in real and synthetic examples.
>
> We believe this change clarifies the scope of our contribution. If the reviewers and the chair feel it is still necessary, We are happy to adjust the title to ``**Asymptotic** Anytime-valid, Bayes-assisted, Prediction-Powered Inference'', although we note that this may become somewhat lengthy.
>
> ### On the Questions for Future Work
>
> __Extension beyond iid data__
>
> Our assumptions mirror those in the standard PPI framework, in which the data are assumed to be i.i.d. The strong coupling results (e.g., Proposition 2) are derived under this assumption. Extending the analysis beyond the i.i.d. case, as suggested by the reviewer, is an interesting research direction that would require careful investigation and an adaptation of the coupling results.
>
> __Update the prior in an online fashion__
>
> In the current method the prior is fixed. Designing an online-adaptive prior that preserves both the tractability of Equation (8) and the asymptotic guarantees appears non-trivial; we regard this as promising future work.

---

### Official Review · Reviewer_qqUX · 2025-07-02

**Clarity:** 3
**Significance:** 3
**Originality:** 3
**Rating:** 5
**Confidence:** 2

**Summary:**

This study extends the framework of Prediction-Powered Inference (PPI) to a sequential setting, introducing methods for constructing anytime-valid confidence sequences that incorporate predictions from black-box models. The approach builds on control variates and leverages the method of mixtures and Ville’s inequality. The authors also develop Bayes-assisted variants that incorporate prior beliefs about prediction quality, aiming to improve the efficiency of inference when predictions are accurate. The methods are evaluated on both synthetic and real data.

**Questions:**

Here are a few points that addressed would help the reader to better evaluate the scope and applicability of the proposed methods:

- The method involves using prior information about the predictions to improve inference. Could the authors give some more intuition or examples for how to choose a good prior in practice? It is not immediately clear how one would decide between different types, and some guidance would be appreciated.

- When the prior is misspecified, the experiments suggest that heavier-tailed priors help. Could this be formalised, or is there a general recommendation for robustness?

- The multivariate setting is discussed in the supplement, but there are no experiments illustrating it. Including one or two practical examples would strengthen the paper.

**Ethical Concerns:**

["NO or VERY MINOR ethics concerns only"]

**Final Justification:**

I believe the authors have appropriately addressed my points, and maintain that this is a strong contribution. I thus keep my recommendation for acceptance.

**Limitations:**

Yes.

**Paper Formatting Concerns:**

NA.

**Quality:**

3

**Strengths And Weaknesses:**

The study addresses the relevant problem of performing valid statistical inference in settings where data arrives over time and predictions from machine learning models are available. The methodological framework is well-motivated and appears theoretically solid, though I did not check the proofs in detail. The extension from fixed-time to time-uniform (anytime-valid) inference is significant, and the use of prior information to tighten intervals seems both novel and potentially impactful.

The presentation is generally clear. However, the technical depth may make the paper challenging to follow for readers not familiar with the literature on confidence sequences or prediction-powered inference. A few parts, particularly around the practical aspects of prior selection and computational cost, could be expanded.

Empirical results support the theoretical claims. That said, the experiments focus on scalar parameters, and while multivariate extensions are discussed in the supplement, no empirical results are shown. Including such results would be important to demonstrate the generality of the approach.

---

> ### Author Rebuttal · Authors · 2025-07-31
>
> We sincerely thank the reviewer for their positive feedback and for recognizing the relevance of our work.
>
> ### About the technical depth of the paper
>
> As mentioned by the reviewer, our presentation is "generally clear", but we acknowledge that our paper is technically deep and that reading may be challenging for a reader not completely familiar to the subject. To address this issue we will add additional background on PPI and Confidence Sequences that should make this paper accessible to a wider audience.
>
> ### About the prior selection
>
> As mentioned in our response to YGZJ, for practical guidance on prior selection, we recommend using priors with heavy tails, as they are more robust to mispecification.
> In particular, the width of the confidence sets induced by polynomial-tailed priors, such as the Student-t, grow logarithmically with the error in the prior mean (see Eq. 8).
> This is in contrast to the Gaussian and Laplace priors, whose confidence sets grow linearly and with the square root, respectively.
> However, the Student-t prior's greater robustness comes at a computational cost, as computing the associated $\eta_t$ requires numerical integration.
> In settings where this is a concern, the Laplace prior might represent a compelling alternative, as it exhibits both closed-form $\eta_t$ and slower confidence set growth rate than the Gaussian prior
>
> ### About the computational cost
>
> As mentioned in Section 7, the main drivers of computational cost for our method are the iteration over a grid of values for $\theta$, which is required for some estimation problems, like for standard PPI, and the computation of $\eta_t$, which is required for the Bayes-assisted methods. In particular, when gridding is required, the numbers of evaluations of $\eta_t$ scales linearly with both the grid size and the number of labelled samples $n$. For heavy-tailed priors, such as the Student-t, where $\eta_t$ is computed via numerical integration, this can become quite expensive. We will make sure to clarify this in the revised manuscript.
>
> ### About our multivariate extension
>
> To illustrate the multivariate extension of our method, we study a multivariate version of the mean estimation task with biased predictions in Section 6.1. In particular, we sample $5$-dimensional $X_i \overset{iid}{\sim} \mathcal{MVN}(\mathbf{0}, \Sigma)$, where $\Sigma \in \mathbb{R}^{5 \times 5}$ is a Toeplitz covariance matrix with entries $\Sigma_{ij} = 0.5^{|i-j|}$, and define $Y_i = X_i + \epsilon_i$, where $\epsilon_i$ is sampled i.i.d.~from a standard normal distribution, so that $\theta^* = \mathbb{E}[Y] = \mathbf{0}$. Then, we proceed as in Section 6.1 and define biased predictions $f(X_i) = X_i + \nu$, where $\nu \in \mathbb{R}$ controls the bias level of the predictor.
>
> The table below compares anytime-valid multivariate classical inference, unassisted PPI, and Bayes-assisted PPI under a Gaussian prior in terms of the volume of the resulting confidence ball at $n = 100$ and $N = \infty$ when varying $\nu$ between $0$ and $5.0$, averaged over $1000$ repetitions.
>
> | $\nu$ | 0.0 | 1.0 | 2.0 | 3.0 | 4.0 | 5.0 |
> |---|---|---|---|---|---|---|
> | classical | 252.6 | 256.4 | 251.3 | 254.3 | 252.7 | 252.3 |
> | PPI | 24.4 | 24.2	| 24.4 | 24.0 | 24.3 | 24.1 |
> | PPI (G) | 0.06 | 0.36 | 3.8 | 21.6 | 85.9 | 241.4 |
>
> The behaviour of the methods considered follows the same pattern as in the scalar case. In particular, unassisted PPI always outperforms classical inference, while maintaining roughly the same volume across all values of $\nu$. On the other hand, Bayes-assisted PPI outperforms unassisted PPI when the predictor is accurate ($\nu = 0,\dots,3$), but eventually suffers from the mispecification of the zero-centered prior when the predictor is poor ($\nu = 4,5$).
>
> While this is not currently possible for the rebuttals, we will include this result in the form of a figure similar to Figure 2 in Section 6.2 to our revised supplementary material.
>
> We believe these changes will address the reviewer's concerns and substantially improve the paper's clarity and impact. We thank the reviewer again for their valuable guidance.

---

> > ### Comment · Reviewer_qqUX · 2025-08-08
> >
> > I thank the authors for appropriately addressing my points. I believe this is a strong contribution and thus keep my recommendation for acceptance.

---

### Official Review · Reviewer_YGZJ · 2025-07-03

**Clarity:** 3
**Significance:** 2
**Originality:** 2
**Rating:** 4
**Confidence:** 3

**Summary:**

This paper extends the Prediction-Powered Inference (PPI) framework, which uses machine learning predictions to improve inference efficiency with limited labeled data, to the sequential (anytime-valid) setting. The authors develop new procedures for constructing confidence sequences that remain valid uniformly over time, as opposed to traditional fixed-time intervals. Their approach incorporates Bayesian prior information about prediction quality to further tighten intervals when appropriate, leveraging Ville’s inequality and the method of mixtures. Experiments on both synthetic and real datasets demonstrate the methods.

**Questions:**

1. **Figure 3 Interpretation**: In Figure 3, all methods show a miscoverage level below 0.1. Could the authors clarify why this is the case? Does this suggest that the proposed and baseline methods are conservative in practice?

2. **Predictor Performance in Real Data**: What is the predictive accuracy of the machine learning model used in the real data experiments? Providing this information would help contextualize the observed efficiency gains of the proposed methods.

3. **Wording on Predictor Accuracy (Line 215)**: The statement, “In many modern applications, extremely accurate black-box predictors are available,” seems overly strong and may not hold in many settings. I recommend revising this claim and/or providing appropriate references to support it.

**Ethical Concerns:**

["NO or VERY MINOR ethics concerns only"]

**Final Justification:**

The authors have adequately addressed my concerns regarding the paper. However, I did not assign a higher score because the work appears to be primarily theoretical, with limited practical applicability. For instance, in many scientific domains, any-time valid inference is not commonly used, and the examples presented in the paper are synthetic rather than drawn from real-world studies. The absence of citations to scientific literature that employs any-time valid inference further limits the perceived relevance. Additionally, applying the method requires making several nontrivial choices, such as selecting appropriate priors, which may present challenges for practitioners.

**Limitations:**

See weakness

**Quality:**

3

**Strengths And Weaknesses:**

Strengths: :
1. The paper tackles a relevant open problem by moving PPI from fixed-time confidence intervals to anytime-valid confidence sequences, for online, streaming, and sequential learning setting.
2. The theory is solid.

Weaknesses:
1. **Motivation and Practical Relevance**: The motivation for this work is not fully convincing. While the authors position their method for online and sequential settings, the real data experiments do not actually involve true online data; rather, the authors simulate an online scenario from static datasets. This leaves open the question of whether the proposed method addresses genuine challenges encountered in real-world online applications. (or just a combination of PPI and any-time valid inference) The connection between practical needs and the proposed methodology could be better articulated and supported with more realistic examples or use cases.
2. **Limited Scope of Empirical Demonstration**: The PPI method is applicable to general M-estimation problems. But the paper only demonstrates their methods (extension of PPI) for mean estimation. This is a significant limitation, as it does not provide evidence of the method’s utility in more complex or widely used models, such as linear or logistic regression. Expanding the empirical evaluation to additional settings would strengthen the impact and applicability of the work.
3. **Dependence on Predictor Quality**: Although prior information can mitigate some issues, the practical efficiency gains over standard approaches still depend heavily on the auxiliary predictor’s accuracy. In practice, guidance for diagnosing or calibrating this prior could be more detailed.

---

> ### Author Rebuttal · Authors · 2025-07-31
>
> We thank the reviewer YGZJ for their thorough review and constructive feedback. We are encouraged that they found our work to be theoretically solid and that it addresses a relevant open problem.
>
> Below, we address the weaknesses and questions raised by the reviewer.
>
> ### On Motivation and Practical Relevance
>
> We thank the reviewer for raising this important point regarding the practical motivation and experimental design. We fully agree that demonstrating real-world utility is essential.
>
> While we use static datasets, these were not chosen arbitrarily. Each one reflects an inherently sequential process: nightly astronomical observations for GALAXY, daily price updates for FLIGHT, and regularly refreshed satellite imagery for FOREST.
>
> Because our method is anytime-valid, it enjoy several desirable properties from inference based on e-values, such as optional stopping. Our experiments focus on evaluating baselines and PPI methods in simulated sequential settings, to allow controlled and transparent performance comparisons. However, anytime-valid methods like ours offer important practical benefits. For instance, they enable practitioners to stop costly data collection as soon as a desired region width is achieved. As clearly shown in our experimental figures, our method can often dramatically lower the stopping time.
>
> Our simulation of sequential data arrival thus faithfully reflects how such data is generated and used in real-world scenarios. This design enables us to rigorously evaluate the core contribution of our work: the construction of anytime-valid confidence sequences that support ongoing monitoring and decision-making as data accumulates. We will ensure that practical use cases are discussed in more detail in the revised manuscript.
>
> ### On the Scope of Empirical Demonstration
>
> We thank the reviewer for the suggestion on how to strengthen the impact of our work. Indeed, our method, like standard PPI, can be applied to any non-degenerate convex estimation task. To show this, as well as to illustrate the performance of our method when the predictor provided is less accurate, we have experimented with the following additional tasks using datasets discussed in [1]:
> - HEALTHCARE ($N = 316215$), whose task is to estimate the parameters of a **logistic regression model**, where the outcome is a health insurance indicator and the covariate is income, and for which the predictor is good.
> - CENSUS ($N = 378091$), whose task is to estimate the parameters of a **linear regression model**, where the outcome is income and the covariate is age, and for which the predictor is poor.
> - GENES ($N = 59150$), whose task is to estimate the **median expression level** of a population of genes, and for which the predictor is poor.
>
> The experimental procedure follows the same design as in Section 6.2. The table below compares anytime-valid classical inference, unassisted PPI, and Bayes-assisted PPI in terms of the width of the resulting confidence regions at $n = 500, 1000, 2000$ labelled samples, averaged over $100$ repetitions.
>
> | $n = 500/1000/2000$ | HEALTHCARE ($\times 10^{-5}$) | CENSUS ($\times 10^2$) | GENES ($\times 10^{0}$) |
> |---|---|---|---|
> | classical | 2.3/1.5/1.0 | 4.89/3.31/2.29 | 2.02/1.33/0.92 |
> | PPI | 2.0/1.3/0.8 | 4.09/2.79/1.92 | 1.64/0.99/0.67 |
> | PPI (G) | 1.5/1.1/0.7 | 5.79/5.17/4.28 | 3.18/3.05/2.89 |
> | PPI (L) | 1.5/1.1/0.7 | 5.28/4.17/3.05 | 2.90/2.50/1.68 |
> | PPI (T) | 1.5/1.1/0.8 | 4.90/3.75/2.71 | 2.30/1.57/1.09 |
>
> In particular, unassisted PPI outperforms classical inference on all datasets, showing the potential benefit of PPI in the presence of correlation between predictions and true labels. On the other hand, Bayes-assisted PPI outperforms unassisted PPI on the HEALTHCARE dataset, where the predictor is accurate, but not on the CENSUS and GENES datasets, for which the zero-centered priors are mispecified. In these cases, the behaviour of the three priors considered (Gaussian, Laplace, and Student-t) differs significantly. In particular, as expected, heavier-tailed priors suffer less from the mispecification, achieving a lower width than the Gaussian prior, which is the most sensitive to the predictor's accuracy.
>
> While this is not currently possible for this rebuttal, we will include these results in the form of figures similar to that in Section 6.2 to our revised supplementary material. We believe that these results clarify the broad applicability of our method and shed light on the practical implications of prediction accuracy and prior choice.
>
> ### On Predictor Dependence and Prior Guidance
>
> The reviewer raises a key practical point about the predictor's influence and the choice of prior.
>
> The fundamental benefit of our method holds as long as the predictor's output is even moderately correlated with the true labels. When a highly accurate predictor is available, our Bayes-Assisted Confidence Sequence offers superior performance. In settings where prior information about the predictor's quality is weak or unavailable, the Non-Assisted version of our method still provides performance gains over standard approaches and serves as a robust default.
>
> For practical guidance on prior selection, we recommend using priors with heavy tails, as they are more robust to mispecification.
> In particular, the width of the confidence sets induced by polynomial-tailed priors, such as the Student-t, grow logarithmically with the error in the prior mean (see Eq. 8).
> This is in contrast to the Gaussian and Laplace priors, whose confidence sets grow linearly and with the square root, respectively.
> However, the Student-t prior's greater robustness comes at a computational cost, as computing the associated $\eta_t$ requires numerical integration.
> In settings where this is a concern, the Laplace prior might represent a compelling alternative, as it exhibits both closed-form $\eta_t$ and slower confidence set growth rate than the Gaussian prior.
>
> ### Responses to Specific Questions
>
> - **Q1: Figure 3 Interpretation:** The reviewer correctly observes that the miscoverage for all methods is below the nominal 0.1 level. This is an expected and desired consequence of guaranteeing anytime-validity. An anytime-valid confidence sequence must ensure coverage for all t. This is a strictly stronger guarantee than fixed-time coverage. To satisfy the "for all time" condition, the intervals at any specific time t must be slightly conservative. We will clarify this in the figure caption.
>
> - **Q2: Predictor Performance in Real Data:** The table below provides the following metrics for all the experiments performed (including the new ones above): normalised RMSE (NRMSE) for the regression models (FLIGHT, CENSUS, GENES) and cross-entropy (CE) for the binary classification models (GALAXY, FOREST, HEALTHCARE).
>
> | FLIGHT | CENSUS | GENES | GALAXY | FOREST | HEALTHCARE |
> |---|---|---|---|---|---|
> | 0.20 | 1.45 | 0.29 | 0.29 | 0.31 | 0.35 |
>
> - **Q3: Wording on Predictor Accuracy (Line 215):** We appreciate the feedback on this statement, we will substantiate the claim by citing prominent examples where highly accurate predictors are available
>
>     - AlphaFold for protein folding: Jumper, John M. et al. “Highly accurate protein structure prediction with AlphaFold.” Nature 596 (2021): 583 - 589.
>
>     - GraphCast for weather forecasting: Lam, Remi et al. “Learning skillful medium-range global weather forecasting.” Science (2023): 1416-1421.
>
>     - Large-scale models for fraud detection: Pozzolo, A. D., Caelen, O., et al. (2015). "Calibrating Probability with Undersampling for Unbalanced Classification". 2015 IEEE Symposium Series on Computational Intelligence and Data Mining (SSCI)
>
>
> We will also re-emphasize that our framework is still beneficial even with moderately accurate predictors and includes a non assisted method for when predictor quality is unknown.
>
>
> We believe these changes will address the reviewer's concerns and substantially improve the paper's clarity and impact. We thank the reviewer again for their valuable guidance.
>
> [1] Anastasios N. Angelopoulos and Stephen Bates and Clara Fannjiang and Michael I. Jordan and Tijana Zrnic. Prediction-Powered Inference. 	arXiv:2301.09633, 2023.

---

> > ### Comment · Reviewer_YGZJ · 2025-08-06
> >
> > I appreciate the reviewers’ detailed responses in the rebuttal, which adequately address my concerns. I will update my score to reflect this.

---

### Note · Authors · 2025-08-13

We would like to thank all reviewers for their thoughtful and supportive feedback. Your insights have provided us with valuable perspectives that will help us refine and improve our work. We are committed to incorporating the discussed changes, and we believe that, thanks to your input, the revised paper will be stronger in clarity, significance, and impact.

---

### Decision · Program_Chairs · 2025-09-17

**Decision:**

Accept (poster)

**Comment:**

This work extends the Prediction-Powered Inference (PPI) framework to the sequential (anytime-valid) setting. The approach incorporates prior information about prediction quality to adjust interval sizes and the approach is validated on both real and synthetic data sets.

Strengths:  The reviewers felt that theory was robust and noted that the theoretical guarantees are valid even under poor-quality/noisy predictors, which increases applicability.

Weaknesses:  The theoretical results can stand on their own, but the experiments are quite limited.  A better set of experiments would bolster the more practical side of this work and demonstrate its claimed wide applicability.  Some reviewers, outside of this subarea, felt that the contributions could be better motivated for the ML community at large.

Overall:  While the experimental results are limited, the reviewers felt that the theoretical contributions were sufficiently novel and widely applicable. The authors provided asymptotic Type I error guarantees in their rebuttal, which further strengthen their contribution.